# Regularizing Brain Age Prediction via Gated Knowledge Distillation

**Yanwu Yang**[1,2]                                          20B952019@STU.HIT.EDU.CN
**XuTao Guo**[1,2]                                            18B952052@STU.HIT.EDU.CN
**Chenfei Ye**[2]                                             CHENFEI.YE@FOXMAIL.COM
**Yang Xiang**[*2]                                            XIANGYANG.HITSZ@GMAIL.COM
**Ting Ma**[*1,2,3,4]                                         TMA@HIT.EDU.CN

[1] *Harbin Institute of Technology at Shenzhen, China*

[2] *Peng Cheng Laboratory, Shenzhen, China*

[3] *Capital Medical University, Beijing, China*

[4] *Xuanwu Hospital Capital Medical University, Beijing, China*

## Abstract

The brain age has been proven a phenotype with relevance to cognitive performance and brain disease. With the development of deep learning, brain age estimation accuracy has been greatly improved. However, such methods may incur over-fitting and suffer from poor generalizations, especially for insufficient brain imaging data. This paper presents a novel regularization method that penalizes the predictive distribution using knowledge distillation and introduces additional knowledge to reinforce the learning process. During knowledge distillation, we propose a gated distillation mechanism to enable the student model to attentively learn key knowledge from the teacher model, given the assumption that the teacher may not always be correct. Moreover, to enhance the capability of knowledge transfer, the hint representation similarity is also adopted to regularize the model training. We evaluate the model by a cohort of 3655 subjects from 4 public datasets, demonstrating that the proposed method improves the prediction performance over several well-established models, where the mean absolute error of the estimated ages is 2.129 years.

**Keywords:** Knowledge distillation, Brain age estimation, Regularization

## 1. Introduction

Aging is shown to have a significant impact on the brain structural changes, following a general decline in cognitive performance and increased risk of neurodegenerative diseases such as Alzheimer's disease (Abbott, 2011) and Parkinson's disease (Reeve et al., 2014). Researches have demonstrated that MRIs could be used to predict chronological age and show that brain age is vital to help improve the detection of early-age neurodegeneration and predict age-related cognitive decline (Cole et al., 2017). It is an essential prerequisite to achieve accurate brain age estimating for quantifying the predicted age difference as a biomarker. Recently, deep learning methods such as Convolution Neural Network (CNN) have been used to predict brain age and have achieved promising results (Cole et al., 2017; Mouches et al., 2021; Ueda et al., 2019; Jónsson et al., 2019). However, these networks may incur overfitting and suffer from poor generalizations, especially on age prediction with

---

[*] Corresponding Author

insufficient data. The aging process could be hardly controlled, and thus the collection of a suitable dataset requires great effort and costs several years. Consequently most available datasets typically contain a limited number of data samples.

Regularization strategies have been proposed to address this issue such as label distribution learning (Geng et al., 2013; Gao et al., 2020; Hu et al., 2019), label smoothing regularization (Müller et al., 2019), deep expectation (Rothe et al., 2015) and so on. The predictive distributions contains the most succinct knowledge of the model, and thus it is effective to regularize the training with predictive distributions (Yun et al., 2020). On this line, an approach called Knowledge Distillation (KD) (Hinton et al., 2015), as an adaptive version of label smoothing regularization, has been investigated widely and has shown to improve generalization performance (Furlanello et al., 2018). Despite its use in model compression, KD penalizes the prediction with a learned, softened, and more "realistic" version of the teacher's output (Yuan et al., 2020; Kim et al., 2021).

In this paper, we propose to revisit KD in the brain age prediction as a regularization that penalizes the predictive distribution by introducing meaningful knowledge from the teacher model. Our contributions can be summarized as follows: (1) We regularize the model learning with the distilled knowledge from a pre-trained teacher model with the same architecture. Additional knowledge of the softened logits output is introduced to reinforce the learning process. (2) We leverage a Gated Distillation (GD) mechanism to guide the model to attentively learn from the teacher model. Since the teacher model might bring unconfident knowledge and give highly erroneous guidance to the student, the transferred knowledge is gated by utilizing the teacher loss as a confidence score. (3) Furthermore, the intermediate-level hint representations learning is adopted to regularize the model to achieve more accurate latent features.

We demonstrate the effectiveness of our proposed method on several well-estimated convolutional neural networks, such as ResNet, DenseNet, SFCN, and DeepBrainNet on a cohort of four public datasets with 3655 subjects with the age range of 16-92 years. In our experiments, the performance on the accuracy of our method is consistently lower than other label paradigm regularization methods, and knowledge distillation-based methods, where the DeepBrainNet achieves the best with a mean absolute error of 2.129 years, a Pearson correlation of 0.987, and a cumulative score of 90.47%. With the regularization method, we won the first in Tencent AIMIS Challenge 2021 in brain age prediction[1]. Our code is available in the Git repository [2].

## 2. Method

### 2.1. Regularization with Knowledge Distillation

Knowledge distillation is a technique to transfer knowledge from a teacher model to a student model, providing extra supervision signals in terms of the neighboring labels similarities learned by the teacher. The student learns from more informative sources and predictive probabilities from the teacher. As is proven in previous studies, knowledge distillation, which can be interpreted as a learned label smoothing regularization (Yuan et al., 2020; Kim

---

1. https://contest.taop.qq.com/

2. https://github.com/podismine/BrainAgeReg

et al., 2021), has the flexibility of rescaling gradients (Tang et al., 2020) and transferring dark knowledge (eg. the knowledge on wrong predictions) in classification tasks (Yun et al., 2020).

### 2.1.1. MULTI-LABEL LEARNING IN BRAIN AGE REGRESSION.

Note that the characteristics of KD mentioned above exist (mostly) in the classification tasks, but not in the regression tasks (Cheng et al., 2018). The regression network predicts unbound, continuous values that are plagued with an unknown error distribution, without access to any dark knowledge. To obtain more informative labels and robust regularization methods, we first convert the single value regression to multi-label learning (MLL) (Tsoumakas and Katakis, 2007; Zhang and Zhang, 2010) problem, posing the brain age regression as a classification task.

MLL quantizes the single values into a multi-label set $L$, which is defined as a fixed label set, interpreting the age range. To reduce the disturbance of boundary missing values, we define the set as $L = (l_k = 12 + \Delta l \cdot k | k = 0, 1, ..., M - 1)$ ranging from 12 to 96, where it is obtained with a bin step of $\Delta l$ and $M = \frac{84}{\Delta l} + 1$ bins. A real number $p_k$ is assigned to each label $l_k$, representing the degree that the corresponding label describes the instance. These labels sum to 1, and the expectation is equal to the real age value $y = \sum_k l_k p_k, p_k \in (0, 1)$.

In our implementation, the bin step $\Delta l$ is set as 4 with the $L$ set of $M = 22$ bins, to reduce the number of labels and mitigate the computational burden. To further describe the label ambiguity between ages, the multi-label learning is extended to distribution learning by generating age labels with a normal distribution with a hyper-parameter $\theta$:

$$q_k = \frac{1}{\sqrt{2\pi}\theta} e^{-\frac{(l_k - y)^2}{2\theta^2}}, p_k = \frac{q_k}{\sum_k q_k} \tag{1}$$

The Kullback-Leibler (KL) divergence is usually used to measure the similarity between the output prediction $\hat{p}(x)$ and the manually designed distribution $p(x)$:

$$L_{DL} = KL(p(x)||\hat{p}(x))) = \sum_{i=0}^{N-1} \sum_{k=0}^{M-1} p_{i,k}(x) log \frac{p_{i,k}(x)}{\hat{p}_{i,k}(x)} \tag{2}$$

where an output probability $\hat{p}_k$ is obtained from the model output $z_k(x)$ with a softmax function $\hat{p}_k = \frac{\exp z_k(x)}{\sum_j \exp z_j(x)}$

### 2.1.2. KNOWLEDGE DISTILLATION

We denote $T$ and $S$ as the teacher and student networks respectively. The teacher model is first trained with the objective function (2) separately, and the student learns from a weighted combination of KL divergence and soft targets from the teacher. Knowledge distillation trains the student model using the KL divergence, and a relaxation temperature $\tau$ is introduced to soften the signal arising from the teacher output:

$$\hat{p}^T(x;\tau) = softmax(z^T(x)/\tau), \hat{p}^S(x;\tau) = softmax(z^S(x)/\tau), \tag{3}$$

The student network is then trained to optimize with the following objective function:

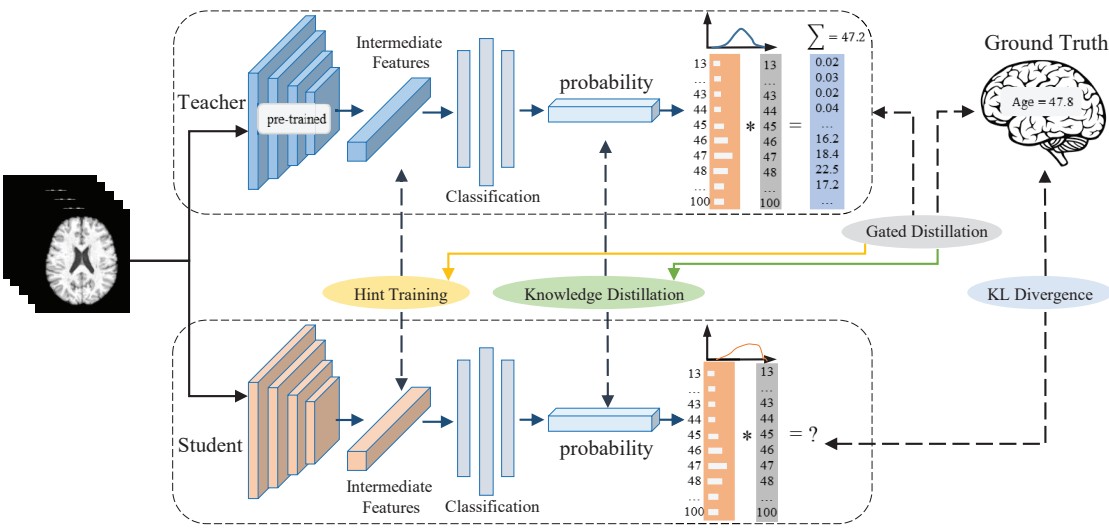

Figure 1: Details of network architecture for teacher (upper) and student training (lower). The implementations of hint training, knowledge distillation, and gated distillation are figured in brief.

$$L_{KD} = KL(\hat{p}^T(x;\tau)||\hat{p}^S(x;\tau)) \tag{4}$$

Finally, the formulation for the student training with KD can be obtained as a regularization form by multiplying a square of the temperature $\tau^2$ with a weight $\lambda$:

$$L = KL(p^S(x)||\hat{p}^S(x)) + \lambda\tau^2 KL(\hat{p}^T(x;\tau)||\hat{p}^S(x;\tau)) \tag{5}$$

Note that the first term in (5) is the KL divergence between the student output and the ground truth, and the second term reinforces the student network to learn from the softened output of the teacher model.

### 2.2. Gated Distillation from the Teacher Model

Considering that teacher models do not always bring good knowledge, we regularize the student model to learn from the teacher model when the teacher provides confident information. To this end, we re-weighted the teacher prediction error as a confidence score to guide the student model for training. Intuitively, when the error is too large to supervise, the student would only learn from the ground truth by itself without learning from the teacher model. The Gated Distillation (GD) mechanism is obtained as:

$$\psi_k = 1 - clip(\frac{||o_k^T - o_k^{GT}||}{\eta}, 1), o_k = \sum_k l_k \hat{q}_k \tag{6}$$

where $\psi_k$ is the transferring weight for $k$-th sample, and $GT$ denotes the ground truth labels. And $o$ denotes the prediction of the models and is obtained with the expectation of

the label distribution. The $\eta$ rescales the prediction error, and the *clip* function restraints the weights as $\psi \in (0,1)$. Different from the Attention Imitation Loss (Saputra et al., 2019), our design aims to reduce the complexity burden, at the same time to limit the model to learn knowledge from the teacher model with specified confidence. The setting of $\eta$ and *clip* operation threshold the prediction error with an upper bound. We consider that the teacher model could not give meaningful knowledge to the student model when the prediction is out of this range. Our implementation helps to mitigate the disturbance of the unconfident transferred knowledge and contributes to a more stable training.

### 2.3. Regularization with Latent Representations

Considering that regularizing the model training with the only output of the models would not obtain equal generalization capability to the teacher model, we introduce the intermediate-level hint training to guide the student for training. The intermediate-level hint training (HT) provides a novel way in KD's transferring knowledge, which achieves success in training the student with deeper or shallow layers (Romero et al., 2014). We also argue that this is also an important regularization for self-knowledge transferring in mimicking the generalization capability of the teacher. The objective function is designed as:

$$L_{HT} = ||h_l^T(x) - h_l^S(x)||^2 \tag{7}$$

where the hint layer is chosen by $l$-th layer. Here, the $L_2$ loss is implemented to minimize the discrepancy between the representations of the teacher and the student models.

Finally, we propose the modification of intermediate-level hint training termed Adaptive Learning with Knowledge Distillation as follows:

$$L = L_{DL} + \lambda_1 \psi L_{KD} + \lambda_2 \psi L_{HT} \tag{8}$$

$$= KL(p^S(x)||\hat{p}^S(x)) + \lambda_1 \tau^2 \psi KL(\hat{p}^T(x;\tau)||\hat{p}^S(x;\tau)) + \lambda_2 \psi ||h_l^T(x) - h_l^S(x)||^2 \tag{9}$$

where the hint representation similarity is combined with the weight $\lambda_2$, and the $\psi$ is the Gated Distillation weight as seen in (6).

## 3. Experiments

### 3.1. Datasets and Preprocessing

The methods were evaluated on T1-weighted MR images from a cohort of four public datasets including the IXI database (http://brain-development.org), the Alzheimer's Disease Neuroimaging Initiative (ADNI) (Jack Jr et al., 2008), the Open Access Series of Imaging Studies (OASIS)(Marcus et al., 2010), and 1000 Functional Connectomes Project (1000-FCP, http://www.nitrc.org/projects/fcon_1000). Only healthy subjects were selected in our experiments, with no indication of neurological pathology, and no psychiatric diagnosis. The ADNI and OASIS datasets are public with longitude studies, where 1024 and 1028 adult and elderly subjects with ages ranging from 42 to 92 are included. The 1000-FCP projects mainly cover the young with 1040 subjects in the mean age of 25, and the IXI dataset covers 563 subjects with a full range of ages. A total of 3655 T1-weighted

MRI images of the subjects aged 16-92 years old are selected to form our cohort. All the images were acquired at either 1.5T or 3T T1-weighed MRI.

All the T1w images were processed including AC-PC aligns, brain skull stripping, bias field correction (Sled et al., 1998), and linear-normalization into the standard MNI space. Additionally, z-score normalization is employed to narrow the gap between different data centers and is shown to improve the synthesis results and is vital for successful deep learning-based MR image synthesis (Reinhold et al., 2019). After preprocessing, all images are down-sampled trilinearly into the standard $2mm^3$ MNI space and padded into the size of $96 \times 112 \times 96$. 5-fold cross-validation was implemented for evaluation.

### 3.2. Performance evaluation

The performance is evaluated by the mean absolute error (MAE), Pearson correlation co-efficient (PCC), and cumulative score (CS). PCC measures the correlation between the predicted ages and the chronological ages. The CS is the accuracy of age estimation within a threshold $\alpha$, which is obtained by: $CS(\alpha) = \frac{N_{e \leq \alpha}}{N} \times 100\%$, where $N_{e \leq \alpha}$ is the number of samples on which the absolute error of prediction $e$ is no higher than the threshold $\alpha$. Moreover, group comparisons involved Wilcoxon test are implemented on the absolute error between the metric regression baseline and other methods, and the calculated p values are used to measure the performance improvement.

### 3.3. Experimental Setup

**Network architecture.** We employ three well-estimated models including ResNet18, ResNet50 (He et al., 2016), and DenseNet121 (Huang et al., 2017). To suit our 3D neu-roimaging data, we utilized the standard architecture and replace the 2D operations with 3D. And a global average pool is applied to average the features. Besides the common structure mentioned before, two neural networks: SFCN (Peng et al., 2019), DeepBrainNet (Bashyam et al., 2020) that are specially designed for brain age estimation are also imple-mented, which are published recently and achieve state-of-the-art performance. The SFCN contains six convolution, batch normalization, activation, and max-pooling layers with the channels of convolution as [32, 64, 128, 256, 256, 88] respectively. The DeepBrainNet is implemented based on the Inception-Res-V2 model. All these models encode the image data into 88 features. A multiple layer perception with three layers is utilized to classify the features into 22 probabilities followed by a softmax function.

**Comparison and ablation studies.** In this paper, we first compared our method with the baselines including metric regression, multi-label learning, and label distribution learning (LDL). LDL can be interpreted as a special case of MLL by setting the label set with a distribution. Moreover, two recently published KD methods that achieve the state-of-the-art performances in classification are also implemented for comparison, including Teacher-free Knowledge Distillation (TF-KD) (Yuan et al., 2020), and Progressive Self-Knowledge Distillation (PS-KD) (Kim et al., 2021). In detail, TF-KD deploys self-training that a model is first pre-trained and then provides a soft label to train itself again. PS-KD transforms the fix pre-trained teacher model to that of the past predictions. Compared with these two methods, our proposed method is built based on TF-KD and attentively refines the distilled knowledge (hint representations and soft labels) using the gating mechanism. Especially,

PS-KD is a one-stage training method, while TF-KD and our proposed methods are within two stages. To further evaluate the effectiveness of our proposed method, we conducted ablation studies on hint learning and Gated Distillation. We searched the hyper-parameters to obtain objective results for all these methods. The multi-label set length in this paper is unified to 22 with a bin step of 4. In KD-based methods, the temperature $\tau$ is decided with a search of [1, 5, 10, 25]. The $\lambda_1$ and $\lambda_2$ are searched within [0.2, 0.4, 0.6, 0.8].

**Training details.** All the models are trained with the same setting. In detail, the networks are trained by the Adam optimizer on the PyTorch 1.6 platform, with an initial learning rate of 1e-6, an L2 weight decay coefficient of 5e-5. The learning rate is increased linearly to 1e-4 in 20 warmup epochs. The best model was obtained based on the validation loss and trained with 300 epochs. The batch size is set to 32 and it takes around 16 hours for training on two NVIDIA V100 GPU with 32G memory. All the models were trained from scratch. To reduce the risk of overfitting, two data argumentation methods were applied during training—random rotation and random shifting. The rotation angles were between $-10^\circ$ and $10^\circ$ and the input was randomly shifted by between -5 and 5 voxels along every axis with equal probability. The hyper-parameter $\theta$ is set as 2, and $\eta$ is 5.

## 4. Results

Table 1 shows the validation accuracy of five well-estimated CNN models in terms of MAE, PCC, CS with $\alpha = 5$, and p value, where the best results of each model is shown in bold. All the results reported are averaged with the 5-fold cross-validation. We compared our proposed method with the metric regression, multi-label learning, label distribution learning, and knowledge distillation-based methods including TF-KD, PS-KD. Several observations can be obtained: (1) Not all the regularization methods achieve improved performance in brain age prediction. For example, label paradigms such as multi-label learning and distribution learning did not perform better than the baseline metric regression on ResNet-18 and ResNet-50. (2) On the other hand, KD-based methods including PS-KD and TF-KD perform better than MLL and DLL. For example, on the ResNet-50, the MAE is 2.440 and 2.459 by TF-KD and PS-KD respectively. And our proposed method even reduced the MAE to 2.395. PS-KD performs worse than TF-KD and our method on five models, which is not consistent with previous studies (Kim et al., 2021). We suspect that these results are caused by the unstable convergence in the brain age prediction task with limited and insufficient data. The teacher model of our method and TF-KD is fixed and pre-trained, however, it is dynamic in PS-KD. (3) The two specifically designed networks for brain age estimation (SFCN and DeepBrainNet) achieve better performance on accuracy than the general-purpose models. The lightweight SFCN network with 6 layers using regularization such as LDL provides better results than deep residual and densely connected networks with more than 18 layers. This is the same as the finding in (Peng et al., 2019) that a lightweight model can achieve even better performance for brain age estimation. (4) Finally our proposed method consistently outperforms other regularization methods on all the five well-estimated models, where the DeepBrainNet achieves the best performance with an MAE of 2.129, a CS of 90.47%, and a PCC of 0.987. Significant improvements are found in SFCN and DeepBrainNet (p = 0.001).

On the other hand, to further inspect the effectiveness of the proposed method, we conducted ablation studies on the gated distillation mechanism and hint training. In Table 2, we observe that the gated distillation plays an important role in improving performance, especially on ResNet-18, ResNet-50, DenseNet-121, and DeepBrainNet. In addition, the hint training slightly improves the performance on all the models except ResNet-50. We suspect that the unsatisfactory results are caused by over-regularization, which is due to an inappropriate selection of the intermediate representations. However, when we combine the gated distillation and hint training, further improvements are achieved on all the models.

Table 1: Performance with different regularization methods on five models.

| | Metric Regression | | | | MLL | | | |
|---|---|---|---|---|---|---|---|---|
| | MAE | CS($\alpha = 5$) | PCC | p value | MAE | CS($\alpha = 5$) | PCC | p value |
| ResNet-18 | 2.642 (0.118) | 84.93%(2.677) | 0.983 (1.032e-4) | - | 2.679 (0.212) | 85.06% (1.918) | 0.983 (9.539e-5) | 0.985 |
| ResNet-50 | 2.489 (0.235) | 87.06% (1.976) | 0.984 (2.459e-4) | - | 2.528 (0.217) | 86.20% (2.585) | 0.984 (1.339e-4) | 0.829 |
| DenseNet-121 | 2.507 (0.167) | 86.49% (1.051) | 0.984 (8.480e-5) | - | 2.512 (0.123) | 87.62% (0.692) | 0.984 (8.281e-5) | 0.949 |
| SFCN | 2.662 (0.110) | 85.35% (0.637) | 0.983 (9.625e-5) | - | 2.441 (0.078) | 87.34% (1.188) | 0.985 (3.101e-4) | 0.063 |
| DeepBrainNet | 2.521 (0.294) | 86.63% (1.3308) | 0.984 (4.525e-4) | - | 2.383 (0.166) | 87.48% (0.813) | 0.985 (6.940e-5) | 0.299 |
| | LDL | | | | PS-KD + LDL | | | |
| | MAE | CS($\alpha = 5$) | PCC | p value | MAE | CS($\alpha = 5$) | PCC | p value |
| ResNet-18 | 2.674 (0.313) | 84.21% (4.182) | 0.983 (1.617e-4) | 0.984 | 2.618 (0.123) | 85.21% (3.522) | 0.983 (9.428e-5) | 0.611 |
| ResNet-50 | 2.510 (0.191) | 86.49% (1.578) | 0.984 (2.270e-5) | 0.853 | 2.459 (0.139) | 86.77% (1.512) | 0.985 (7.634e-5) | 0.783 |
| DenseNet-121 | 2.454 (0.184) | 87.05% (1.500) | 0.985 (8.212e-5) | 0.614 | 2.446 (0.075) | 86.77% (2.810) | 0.985 (1.624e-4) | 0.552 |
| SFCN | 2.345 (0.064) | 87.48% (0.316) | 0.986 (6.390e-5) | 0.005* | 2.388 (0.165) | 87.20% (1.253) | 0.985 (7.082e-5) | 0.052 |
| DeepBrainNet | 2.349 (0.134) | 87.62% (1.797) | 0.986 (1.352e-4) | 0.030* | 2.364 (0.034) | 86.20% (0.384) | 0.985 (6.810e-5) | 0.011* |
| | TF-KD + LDL | | | | Ours | | | |
| | MAE | CS($\alpha = 5$) | PCC | p value | MAE | CS($\alpha = 5$) | PCC | p value |
| ResNet-18 | 2.606 (0.199) | 85.20% (2.515) | 0.983 (1.978e-4) | 0.631 | **2.405 (0.169)** | **87.05% (1.471)** | **0.985 (7.494e-5)** | 0.096 |
| ResNet-50 | 2.440 (0.175) | 87.78% (1.439) | 0.985 (7.552e-5) | 0.747 | **2.395 (0.157)** | 86.77% (3.504) | **0.985 (7.179e-5)** | 0.538 |
| DenseNet-121 | 2.435 (0.134) | 87.06% (2.457) | 0.985 (6.963e-5) | 0.432 | **2.341 (0.084** | **88.90% (1.371)** | **0.986 (1.516e-4)** | 0.021* |
| SFCN | 2.332 (0.040) | 87.62% (0.416) | 0.986 (2.697e-5) | 0.003** | **2.269 (0.036)** | **89.76% (0.773)** | **0.986 (6.462e-5)** | **0.001**** |
| DeepBrainNet | 2.338 (0.061) | 87.20% (0.380) | 0.986 (6.531e-5) | 0.008** | **2.129 (0.026)** | **90.47% (0.267)** | **0.987 (3.098e-5)** | **0.001**** |

Table 2: Ablation studies on adaptive transferring weights and hint learning.

| | Ours w/o GD | | | | Ours w/o HT | | | |
|---|---|---|---|---|---|---|---|---|
| | MAE | CS($\alpha = 5$) | PCC | p value | MAE | CS($\alpha = 5$) | PCC | p value |
| ResNet-18 | 2.515 (0.199) | 87.06% (1.412) | 0.983 (8.813e-5) | 0.383 | 2.442 (0.181) | 87.62% (2.409) | 0.985 (7.439e-5) | 0.283 |
| ResNet-50 | 2.527 (0.161) | 86.20% (2.541) | 0.983 (7.677e-5) | 0.829 | 2.427 (0.150) | 86.91% (3.519) | 0.985 (9.291e-5) | 0.638 |
| DenseNet-121 | 2.422 (0.078) | 87.91% (1.398) | 0.987 (4.994e-5) | 0.332 | 2.422 (0.076) | 85.78% (2.498) | 0.985 (7.192e-5) | 0.341 |
| SFCN | 2.441 (0.033) | 88.76% (1.360) | 0.986 (5.901e-5) | 0.098 | 2.332 (0.133) | 87.48% (1.419) | 0.987 (3.883e-5) | 0.004** |
| DeepBrainNet | 2.356 (0.028) | 87.06% (0.409) | 0.986 (2.891e-5) | 0.042* | 2.155 (0.061) | 89.76% (0.815) | 0.987 (1.897e-5) | 0.001** |

## 5. Discussion and Conclusion

In this paper, we demonstrate a novel regularization method based on knowledge distillation and compare it with other regularization methods. These methods pose the brain age regression as a deep classification task, at the same time regularizing the models to learn with calibration and prevent over-confidence. Moreover, our proposed method estimates the brain age by taking the confidence of the teacher knowledge into consideration. This strategy encourages the student to attentively learn with meaningful knowledge. Finally, our method achieves consistent improvement on five well-estimated CNN models.

In our implementation, the brain age prediction is transformed to a classification task instead of directly regressing. On the one hand, this can improve the diversity of labels by extending a single value into a multi-label set, on the other hand, preserving the advantages of KD eg. rescaling gradients, and transferring dark knowledge. The comparison results of the regression models with KD and more detailed evaluation about the results are demonstrated in the appendix.

## Acknowledgments

This study is supported by grants from the Innovation Team and Talents Cultivation Program of National Administration of Traditional Chinese Medicine (NO:ZYYCXTD-C-202004), Basic Research Foundation of Shenzhen Science and Technology Stable Support Program (GXWD20201230155427003-20200822115709001), and the National Natural Science Foundation of China (62106113 and 62106115).

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

## Appendix A. Study of the parameters of $\lambda_1$, $\lambda_2$, and $\tau$

Considering that a full grid search for all these hyper-parameters costs numerous time and resources. We first fix the $\lambda_1$ and $\lambda_2$ and search for the relaxation temperature (the best results are achieved with a $\tau = 10$ on ResNet-18, ResNet-50, and SFCN, and $\tau = 25$ on DeepBrainNet and DenseNet-121). In our comparison, it is found that the value of the temperature is more sensitive to the results than $\lambda_1$ and $\lambda_2$. After that, a grid search over $\lambda_1$ and $\lambda_2$ is carried out. Notably, $\lambda_1 = 0.8$ and $\lambda_2 = 0.8$ are the optimal values.

## Appendix B. Learning curves of different models

As is shown in Fig.2, the L1 loss values of the training and validation on the teacher model and student model using our proposed method are figured in black, grey, green, and blue respectively, and the absolute differences of the training and validation performances are plotted under the learning curves, where the orange describes the differences on the teacher

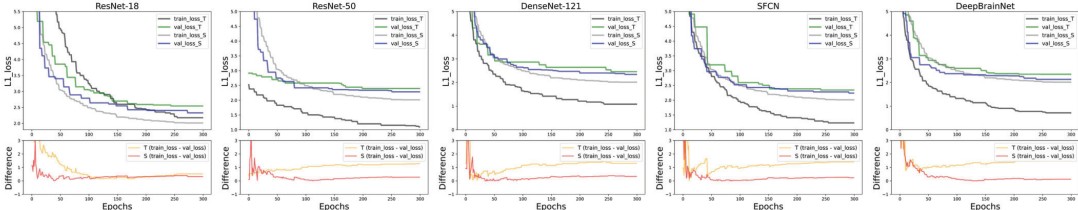

Figure 2: Learning curves with different CNN models based on our proposed method.

model and the red shows those on the student model. To smooth the curve, only the lowest loss values are plotted. The student model got convergence more quickly than the teacher model except for the ResNet-50 model. And the differences in the training and validation performance are reduced on the student model compared with the teacher model.

## Appendix C. Experimental results on regression

In addition to posing the brain age regression as a classification problem, we also compare the results of our method on regression. The experiments are conducted based on the baseline metric regression method, where a two-stage training is carried out by regularizing the model with the Gated Distillation and the hint training.

As is shown in Table 3, our method achieves comparable results with TF-KD, and PS-KD, where slight improvements are obtained on DenseNet-121, and ResNet-50. Attention should be paid to the DeepBrainNet that the three models achieve comparable results, where PS-KD seems more powerful with only a one-stage training. In addition, it is interesting that our proposed method achieves improvement, even slightly, indicating that it might work in regression problems. Overall, our proposed strategies help to increase the accuracy of the brain age prediction in both regression and classification, where the multi-label learning ( and label distribution learning) with KD regularization achieves better performance on accuracy than metric regression with KD.

Table 3: Comparisons with the KD-based methods on regression.

| | Metric Regression | | | PS-KD | | |
|---|---|---|---|---|---|---|
| | MAE | CS($\alpha = 5$) | PCC | MAE | CS($\alpha = 5$) | PCC |
| ResNet-18 | 2.642 (0.118) | 84.93%(2.677) | 0.983 (1.032e-4) | **2.523 (0.237)** | **86.20% (1.938)** | **0.984 (1.533e-5)** |
| ResNet-50 | 2.489 (0.235) | 87.06% (1.976) | 0.984 (2.459e-4) | 2.575 (0.301) | 87.62% (1.565) | 0.986 (4.898e-5) |
| DenseNet-121 | 2.507 (0.167) | 86.49% (1.051) | 0.984 (8.480e-5) | 2.372 (0.109) | 87.06% (2.901) | 0.986 (1.303e-4) |
| SFCN | 2.662 (0.110) | 85.35% (0.637) | 0.983 (9.625e-5) | **2.308 (0.083)** | **87.62% (1.039)** | **0.987 (3.202e-4)** |
| DeepBrainNet | 2.521 (0.294) | 86.63% (1.3308) | 0.984 (2.525e-4) | 2.245 (0.033) | 88.90% (1.298) | 0.988 (3.678e-4) |
| | TF-KD | | | Ours | | |
| | MAE | CS($\alpha = 5$) | PCC | MAE | CS($\alpha = 5$) | PCC |
| ResNet-18 | 2.650 (0.321) | 84.92% (0.654) | 0.982 (8.982e-5) | 2.620 (0.231) | 86.06% (1.036) | 0.982 (4.098e-5) |
| ResNet-50 | 2.775 (0.144) | 84.92% (1.082) | 0.983 (1.190e-4) | **2.432 (0.165)** | **88.19% (0.737)** | **0.985 (6.110e-5)** |
| DenseNet-121 | 2.446 (0.129) | 86.77% (1.003) | 0.985 (6.090e-5) | **2.338 (0.027)** | **87.77% (1.882)** | **0.985 (8.717e-5)** |
| SFCN | 2.601 (0.098) | 85.35% (0.320) | 0.986 (1.324e-5) | 2.373 (0.312 ) | 87.77% (1.109) | 0.984 (1.221e-4) |
| DeepBrainNet | 2.253 (0.043) | 88.62% (1.111) | 0.987 (6.909e-5) | **2.241 (0.130)** | **89.42% (1.991)** | **0.987 (3.676e-5)** |

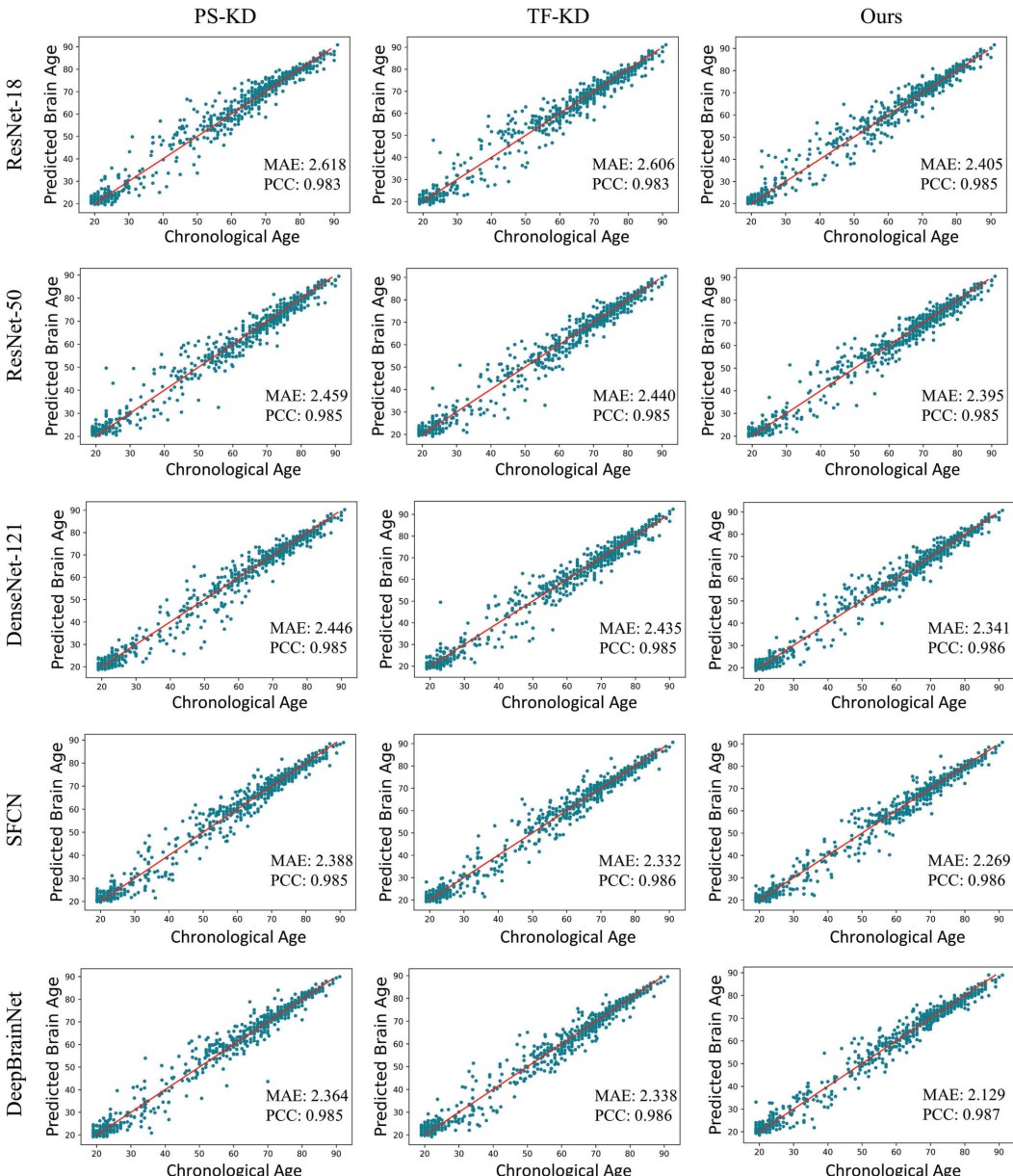

Figure 3: The scatter plots of the predicted brain ages and chronological ages on the three KD-based models: PS-KD, TF-KD, and our proposed model.

## Appendix D. Scatter plots of different models

The scatter plots between chronological age (X-axis) and predicted brain age (Y-axis) of five well estimated CNN models with different regularization methods are shown in Fig. 3.

## Appendix E. Performance comparison with different KD-based methods on each dataset in terms of Mean Absolute Error

The performances on different datasets are shown in Table 4. Our proposed method achieved the lowest MAEs in most cases. Overall, our method is generalizable to different datasets from different sites.

Table 4: Performance of each dataset in terms of MAE.

|  | ADNI | | | OASIS | | |
|---|---|---|---|---|---|---|
|  | PS-KD | TF-KD | Ours | PS-KD | TF-KD | Ours |
| ResNet-18 | 1.315 (0.151) | 1.248 (0.181) | **1.180 (0.129)** | 2.335 (0.121) | 2.389 (0.193) | **2.056 (0.094)** |
| ResNet-50 | **1.161 (0.180)** | 1.439 (0.139) | 1.195 (0.137) | 2.259 (0.163) | **2.237 (0.201)** | 2.246 (0.134) |
| DenseNet-121 | 1.355 (0.217) | 1.244 (0.131) | **1.162 (0.125)** | 2.232 (0.084) | 2.351 (0.164) | **2.060 (0.033)** |
| SFCN | 1.452 (0.092) | 1.360 (0.128) | **1.220 (0.033)** | 2.114 (0.055) | **2.102 (0.032)** | 2.142 (0.127) |
| DeepBrainNet | 1.176 (0.135) | 1.147 (0.089) | **1.009 (0.033)** | 2.441 (0.218) | 2.161 (0.083) | **2.028 (0.028)** |
|  | IXI | | | 1000-FCP | | |
|  | PS-KD | TF-KD | Ours | PS-KD | TF-KD | Ours |
| ResNet-18 | 4.533 (0.334) | 4.486 (0.238) | **4.159 (0.254)** | 3.632 (0.223) | 3.432 (0.158) | **3.275 (0.192)** |
| ResNet-50 | **3.851 (0.071)** | 4.059 (0.137) | 3.880 (0.191) | 3.428 (0.191) | 3.158 (0.137) | **2.964 (0.082)** |
| DenseNet-121 | 4.042 (0.318) | 4.286 (0.313) | **3.994 (0.229)** | 3.107 (0.153) | **2.909 (0.238)** | 3.140 (0.238) |
| SFCN | 3.696 (0.055) | 3.636 (0.171) | **3.408 (0.046)** | 3.033 (0.109) | **2.968 (0.059)** | 3.042 (0.170) |
| DeepBrainNet | 3.861 (0.299) | 3.619 (0.178) | **3.563 (0.144)** | 2.895 (0.029) | 3.064 (0.448) | **2.793 (0.267)** |

