# OpenReview forum: "Regularizing Brain Age Prediction via Gated Knowledge Distillation"
_MIDL.io/2022/Conference — MIDL 2022_

### Official Review · Reviewer_KqAA · 2022-01-20

**Confidence:** 3
**Preliminary Rating:** 3
**Recommendation:** Oral, Poster

**Summary:**

The paper proposes a gated knowledge distillation network for brain age prediction. The experiments are performed on a combination of four datasets. The results show that the proposed method indeed gives better results compared to its counterparts. The method won 1st price in a brain age prediction competition, primarily designed for users in China. This shows that the proposed method indeed can give good brain age prediction results. Overall, the paper proposes a lot of methodological tweaks for knowledge distillation. The majority of these tweaks are justified with an ablation study.

**Strengths:**

* The paper proposes a knowledge distillation framework for brain age prediction. The method is well validated on publicly available datasets.
* Method section of the paper is well written, with the majority of things easy to understand.
* Ablation study and comparison against previously published work shows the usefulness of the method.
* Experiments using multiple different Neural Network architectures is commendable.

**Weaknesses:**

* The section regarding converting the age-regression problem to Distributed learning was unclear. Maybe a graphical representation in the appendix would help in improving clarity.
* The proposed method employs Label Distribution Learning (LDL) + Knowledge Distillation (KD) + Gated Distribution (GD) + Hint Training (HT). Although an ablation study of individual LDL and pair of KD, GD, and HT is provided, it is not clear the usefulness of KD alone.
* In the paper, it is not mentioned if, for the ablation study, LDL was used or not.
* Comparison against the teacher model is not provided; this would help understand the necessity of KD.
* No details of the methodology of KD techniques like TF-KD and PS-KD are provided. A clear sentence highlighting the difference between these methods and the proposed knowledge distillation method would be helpful. This could be provided either in the Introduction/Literature-Review section or during the Experimental setup section.
* The main reasoning behind the proposed regularization method is to improve the generalizability and reduce overfitting. It would be helpful if the authors provided differences in training and validation data performance. This would help analyze if the proposed method improved the generalizability.

**Deanonymize Review:**

no

**Detailed Comments:**

* One important hyper-parameter of the proposed method is $\theta$. In the paper or in the appendix, no mention of the chosen value of this is provided.
* Similarly, the value of hyper-parameter $\eta$ in Eq. 6 is also not mentioned in the implementation detail
* It is not clear why multiplying the temperature parameter $\tau$ with the loss function in Eq. 5 is necessary. .
* Statistical significance analysis is missing in the paper. This is necessary as the difference in PCC between different methods is negligible.
* The paper uses a combination of 4 different publicly available datasets, where each dataset has different age distribution. Maybe reporting the performance of the proposed method for each of these datasets separately would be helpful.
* A graph depicting actual ages on X-Axis and predicted ages on Y-Axis would be more helpful in understanding the performance of the proposed method.
* Methods like SFCN and BrainNetCNN use a type of MLL (converting regression problem to classification with a bin of 1 year) or transfer learning. It is not clear if the authors used a similar type of methodology when comparing their proposed method to these methods or they just used those architectures. A clarification regarding this would be helpful.
* The introduction section needs a little bit better English as there are some typos and some things are not that clear. Ex. Page-2, Para-2, Sentence-3: "To regularize the predictive distributions provides an effective way of learning from insufficient data, for it contains the most succinct knowledge of the model"

**Final Rating After The Rebuttal:**

5: Strong Accept

**Justification Of The Final Rating:**

The authors addressed the majority of my comments. The provided response helps in clarifying all my concerns about the paper. With this in mind, I would change my recommendation to Strong accept with the option of Oral presentation for MIDL 2022.

**Paper Type:**

both

**Questions To Address In The Rebuttal:**

Mainly all points in the weakness section except point 1.  This is necessary as those points raise doubt regarding the usefulness of the proposed method and could help in improving the paper. Overall seems to propose a lot of modification, but many things are not clear.

**Special Issue:**

no

---

### Official Review · Reviewer_LvbD · 2022-01-21

**Confidence:** 4
**Preliminary Rating:** 5
**Recommendation:** Oral

**Summary:**

The manuscript “Regularizing Brain Age Prediction via Gated Knowledge Distillation” by Yang et al. proposes a novel method of regularization for predicting brain age using CNNs. The authors state that many novel approaches to predicting brain age with deep learning suffer from overfitting and poor generalization, typically due to a lack of available data. Regularization techniques have been proposed to address this issue, including knowledge distillation. The authors aim to outperform state of the art in brain age prediction using their novel methods. These methods include the application of KD to brain age prediction with a novel gated distillation (GD) mechanism that utilizes the teacher loss as a confidence score to penalize unconfident guidance. Hint representation learning is also applied to further regularize the model.

**Strengths:**

The paper is well written, the approach is novel and applied to brain age prediction which has been a topic of high interest recently. The proposed method is well described and references to past work are included. The validation of the method is extensive, and the authors used recent state-of-art model architectures specifically developed for brain age prediction. The results are promising.

**Weaknesses:**

Overall, some statements are unclear and could be rephrased/expanded upon to help the reader further understand the topic; and the comparison between the different models could include more statistical validation.

Major comments:

* It would be helpful for the reader to add one sentence summarizing the differences between PS-KD, TF-KD and the proposed approach, especially to better understand the differences we see in the results. The abbreviations (Teacher-free Knowledge Distillation and progressive self-knowledge distillation) should also be defined.
* Sec. 3.3: Training details: Are the parameters (learning rate/decay) the same for all models (ResNETs/ DenseNet /SFCN/DeepBrainNet)?
* Results: Adding std across folds in the tables would help evaluate the differences between models. Furthermore, are the differences between model performances statistically significant? Many of the PCC scores are the same of differ by 0.001, MAEs differ by ~0.02, and percentages differ by less than 1%. Is there any test for significance that can be applied to strengthen the conclusion that the proposed method outperforms the others?
* It would be interesting to see some learning curves of the student models in the appendix.

**Deanonymize Review:**

no

**Detailed Comments:**

Minor comments:

* Abstract - “Mean absolute error of the estimated ages is reduced to 2.129 years.” Reduced from what?
* Sec. 1 - End of page 1-start of page 2: “these networks may incur overfitting and…” Is overfitting the main concern for these methods?
* Sec. 1 - The papers referenced as regularization strategies do not all deal with brain age estimation (e.g., Geng et al., Mueller et al. are for facial age estimation) but the way the text is written makes it sound as if they do. Perhaps consider rewording this statement to acknowledge that these have been applied to different tasks
* Sec. 2.1, start of page 3: explain what dark knowledge is? As it is referenced again in section 2.1.1.
* Sec 2.2: Phrasing at the start of section 2.2: “considering that teacher models do not always bring good knowledge” is slightly unclear. Does this refer to the teacher model predicting the wrong age? What is considered as “good knowledge”? Perhaps consider rewording to better reflect the intended message. Additionally, what is considered an error that is “too large to supervise”?
*  Sec. 2.2 - we re-weighted the teacher prediction error as a confidence sore -> should be “score”
*  Sec. 2.3: “Considering that regularizing the model training with the only output of the models would not obtain equal generalization capability…” is slightly unclear as to what is meant by “equal generalization capability”
*  Sec. 3.3 - PSKD should be “PS-KD”
*  Sec. 4 - the gated distillation plays an more important -> should “a more important”

**Final Rating After The Rebuttal:**

5: Strong Accept

**Justification Of The Final Rating:**

I appreciate that the authors made efforts to integrate reviewers suggestions, which greatly improved the quality of the paper. I am sure that the novelty of the proposed approach will be of high interest for the MIDL attendants.

**Paper Type:**

both

**Questions To Address In The Rebuttal:**

Please see major comments in the "weaknesses" section. Overall, the paper would improve by explaining a little bit more some terms (PS-KD, TF-KD, dark knowledge) and improving the comparison between the different models.

**Special Issue:**

yes

---

### Official Review · Reviewer_tGce · 2022-01-22

**Confidence:** 5
**Preliminary Rating:** 4
**Recommendation:** Poster

**Summary:**

This paper proposes regularization method that penalizes the predictive distribution using knowledge distillation and introduces additional knowledge to reinforce the learning process. A a gated distillation mechanism to enable the student model to attentively learn meaningful knowledge from the teacher model is proposed. The method is evaluated on a cohort of 3655 subjects from four public datasets with ages range of 16-92. The presented mean absolute error of the estimated ages is 2.129 years with their method.

**Strengths:**

- brain age estimation can be important for estimating brain disease and age-related effects
- the combination of methods is interesting
- the experimentation seems appropriate
- results are good
- the paper is reasonably well written
- the gated distillation is an interesting addition

**Weaknesses:**

- potentially limited improvements over naive methods but this might be dataset related
- the overall execution of the paper is sound, despite the averaging of the cross-validation results without providing confidence intervals or significance analysis.

**Deanonymize Review:**

no

**Detailed Comments:**

The only real remark I have is that the results don't really look significant. This might well be a result of having reached the maximum learnable information content of these datasets. FOr example the worst MAE of the most naive application of a ResNet is 2.674, their result is 2.129. How is this relevant in the clinical practice. I don't see how this relativity small error would have a diagnostic downstream effect.
Furthermore it would be interesting to see confidence intervals instead of averaging the results from the used 5-fold cross-validation. This would also allow better significance analysis.


**Final Rating After The Rebuttal:**

4: Weak Accept

**Justification Of The Final Rating:**

Thank you for addressing the points raised above and adding clarifications to the presented work. I have no further comments.
Thank you for addressing the points raised above and adding clarifications to the presented work. I have no further comments.

**Paper Type:**

methodological development

**Questions To Address In The Rebuttal:**


Could you please comment on the maximum acceptable error for brain age estimation in a clinical diagnostic context?

Can it be expected that the minimum error for this task goes towards zero, given the limited sample size and image resolution in the used datasets?


**Special Issue:**

no

---

### Meta-Review · Area_Chair_i5Bg · 2022-02-19

**Recommendation:** Accept (Oral)
**Confidence:** 5

**Metareview:**

There was strong agreement among reviewers recommending acceptance. While some points in initial submission were unclear, the authors responded to these points to the satisfaction of all reviewers. Strengths of the paper include 1) novelty of the proposed approach for regularizing model training with gated knowledge distillation, 2) extensive experimental validation (multiple datasets and network structures / settings), 3) strong results, and 4) good writing.

---

### Decision · Program_Chairs · 2022-02-28

Accept